# KeyVID: Keyframe-Aware Video Diffusion for Audio-Synchronized Visual Animation

## Abstract

Generating video from various conditions, such as text, image, and audio, enables precise spatial and temporal control, leading to high-quality generation results. Most existing audio-to-visual animation models rely on uniformly sampled frames from video clips. Such a uniform sampling strategy often fails to capture key audio-visual moments in videos with dramatic motions, causing unsmooth motion transitions and audio-visual misalignment. To address these limitations, we introduce **KeyVID**, a keyframe-aware audio-to-visual animation framework that adaptively prioritizes the generation of keyframes in audio signals to improve the generation quality. Guided by the input audio signals, KeyVID first localizes and generates the corresponding visual keyframes that contain highly dynamic motions. The remaining frames are then synthesized using a motion interpolation module, effectively reconstructing the full video sequence. This design enables the generation of high frame-rate videos that faithfully align with audio dynamics, while avoiding the cost of directly training with all frames at a high frame rate. Through extensive experiments, we demonstrate that KeyVID significantly improves audio-video synchronization and video quality across multiple datasets, particularly for highly dynamic motions. The anonymous code link for review is `https://anonymous.4open.science/r/KeyVID-73E6`.

## 1 Introduction

Recent years have witnessed remarkable progress in video generation, driven by advancements in diffusion-based models (Xing et al., 2024; Chen et al., 2023a; 2024; He et al., 2022; Singer et al., 2023; Ho et al., 2022b; Guo et al., 2024; Hong et al., 2022; Yang et al., 2024; Fan et al., 2025; Blattmann et al., 2023a;b). These frameworks typically condition the generation process on *text* prompts and/or *image* inputs, where the text provides semantic guidance (*e.g.*, actions, objects, or stylistic cues), while the image specifies spatial composition (*e.g.*, object layout, scene structure or visual styles). Despite their success, these methods largely focus on aligning visual outputs with static text or images, leaving dynamic, time-sensitive modalities such as *audio* underexplored.

Audio-Synchronized Visual Animation (ASVA) (Zhang et al., 2024b) aims to animate a static image into a video with objects' motion dynamics that are semantically aligned and temporally synchronized with the input audio. It utilizes audio cues to provide more fine-grained semantic and temporal control for video generation, which requires deep understanding of audio semantics, audio-visual correlations, and object dynamics. To achieve precise audio-visual synchronization in ASVA, it is crucial to align key visual actions accurately with their corresponding audio signals. For example, given an audio clip of hammering sounds, the hammer in the video should strike the nail exactly when the impact sound occurs. However, this synchronization is constrained by the frame rates of the video generation models. For example, AVSyncD (Zhang et al., 2024b) is trained to generate videos at 6 FPS, posing a significant challenge for audio-synchronized video generation. Since audio carries fine-grained temporal information, the key moments in the audio can be lost in uniformly sampled low frame rate videos (see Fig. 1(a)), leading to compromised audio-video synchronization.

A straightforward solution is to train a video generation model on high frame rate data to match the fine-grained temporal information in audio. However, this brute-force approach treats all time steps equally and introduces redundant frames in low-motion regions. It also fails to leverage the structural information in the

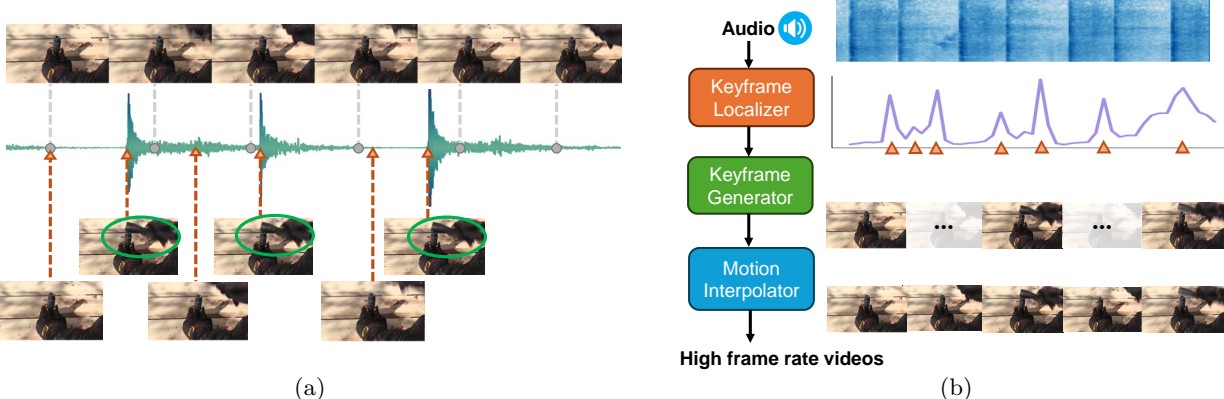

(a)                                                    (b)

Figure 1: (a) **Uniform frames vs. keyframes.** *Top*: Uniformly sampled sparse frames, which fail to capture the key moments evident in the corresponding audio (*Middle*). *Bottom*: Keyframes precisely aligned with the hammer striking down, matching the critical moments in the audio waveform. (b) **KeyVID video generation pipeline.** KeyVID first detects keyframe time steps from the audio input with the *keyframe localizer* and then utilizes a *keyframe generator* to generate the corresponding visual keyframes. Intermediate frames are generated with the *motion interpolator*.

input audio to focus the model capacity on salient moments, which is crucial for audio-visual synchronization. In addition, this approach incurs substantial computational costs in terms of GPU memory and training time. To alleviate this, a two-stage strategy has been proposed that first generates low frame rate videos and then applies frame interpolation to obtain high frame rate videos (Blattmann et al., 2023a; Singer et al., 2023; Ho et al., 2022a). And a random frame rate strategy is proposed to use random frame sampling rates while maintaining a small, fixed number of frames during training (Singer et al., 2023; Zhou et al., 2022). However, the two-stage approach struggles in modeling highly dynamic sequences, where critical events may be lost due to the sparsity of the initial uniform frames, and the random frame rate strategy fails to model long-term temporal dependency at high frame rates due to the limited number of total frames.

In this work, instead of sampling uniform frames, we propose **KeyVID**, a **Key**frame-aware **VI**deo **D**iffusion framework that adaptively selects and generates sparse yet informative keyframes guided by audio cues to capture critical audio-visual events (Fig. 1(b)). We first develop a keyframe selection strategy that identifies critical moments in the video sequence based on an optical flow-based motion score. We train a *keyframe localizer* that predicts such keyframe positions directly from the input audio cue. Next, instead of applying uniform downsampling to video frames, we select the keyframes to train a *keyframe generator*. The keyframe generator explicitly captures crucial moments of dynamic motion that might otherwise be missed with uniform sampling without requiring an excessively high number of frames. Then, we train a specialized *motion interpolator* to synthesize intermediate frames between the keyframes to generate high frame rate videos. The motion interpolator ensures smooth motion transition and precise audio-visual synchronization throughout the sequence. This approach is similar to how the animation industry creates smooth and dynamic movements, where the *Key Animator* establishes key moments in a scene and the *Inbetweener* fills in the gaps to ensure that the movements appear seamless and fluid. This selective temporal focus enables smoother motion transitions and sharper audio-visual synchronization without the overhead of dense uniform sampling.

We conducted extensive experiments across diverse datasets featuring varying degrees of motion dynamics and audio-visual synchronization. We demonstrate that our keyframe-aware approach outperforms state-of-the-art methods in video generation quality and audio-video synchronization. In particular, on the AVSync15 dataset (Zhang et al., 2024b), we achieve an FVD score (Unterthiner et al., 2018) of 263.3, and a RelSync score (Zhang et al., 2024b) of 49.06, outperforming the state-of-the-art by absolute margins of **85.8**, and **3.54**, respectively. Our user study demonstrates a clear preference towards videos generated by KeyVID over those produced by baseline methods.

The main contributions of our work are as follows:

- We propose a novel keyframe-aware audio-to-visual animation framework that first localizes keyframe positions from the input audio and then generates the corresponding video keyframes using a diffusion model.

- We design a keyframe generator network that selectively produces sparse keyframes from the input image and audio, effectively capturing crucial motion dynamics.

- Comprehensive experiments demonstrate our superior performance in audio-synchronized video generation, particularly in highly dynamic scenes with distinct audio-visual events.

## 2 Related Work

**Video Diffusion Models.** Diffusion models (Xing et al., 2024; Chen et al., 2023a; 2024; He et al., 2022; Singer et al., 2023; Ho et al., 2022b; Guo et al., 2024; Hong et al., 2022; Yang et al., 2024; Fan et al., 2025; Blattmann et al., 2023a;b) emerge as powerful tools to generate high-quality videos. For the data sample $\mathbf{x}_0 \sim p_{\text{data}}(\mathbf{x})$, Gaussian noise is added over $T$ steps, creating a noisy version $\mathbf{x}_T$. A model $\epsilon_\theta$ is trained to invert this process by predicting and subtracting the noise. For latent video generation (Xing et al., 2024; Zhang et al., 2023; He et al., 2022; Blattmann et al., 2023b), $\mathbf{x}$ is encoded into a latent vector $\mathbf{z}$ using an encoder $\mathcal{E}(\cdot)$ to reduce computation. The noise-adding diffusion process and the learned reverse process are conducted on $\mathbf{z}$ instead. Recent advancements in video diffusion models leverage pre-trained text encoders (Radford et al., 2021; Raffel et al., 2020) to inject text conditions into the denoising process for text-to-video generations (Blattmann et al., 2023b; Hong et al., 2022; Chen et al., 2023a; Luo et al., 2023). Moreover, image conditioning can also be introduced to enhance video generation by providing visual features that control the visual contents (Wu et al., 2024a; Yang et al., 2023; Li et al., 2023b; Chen et al., 2023b; Wei et al., 2023) or frame conditions (Xing et al., 2024; Chen et al., 2024; Guo et al., 2024; Zhang et al., 2020; Voleti et al., 2022; Franceschi et al., 2020; Babaeizadeh et al., 2018).

**Audio-to-Video Generation.** Compared to text and image, audio provides not only semantic cues but also fine-grained temporal signals for motion generation. Prior studies explored domain-specific audio-conditioned motion synthesis in 2D and 3D (Sun et al., 2023; Zhang et al., 2024a; Wu et al., 2024b; Sung-Bin et al., 2024; Richard et al., 2023), and more recent works leverage pretrained audio encoders (Girdhar et al., 2023; Elizalde et al., 2023) for general video generation. Existing methods either treat audio as a *global feature* for style/semantic control (Hertz et al., 2023; Kim et al., 2023; Wu et al., 2023) or enforce *uniform temporal alignment* with audio clips (Lee et al., 2022; Ruan et al., 2023; Zhang et al., 2024b). However, their motion quality is often limited by low frame rates or costly uniform sampling strategies, especially in highly dynamic scenes. In contrast, we introduces a *keyframe-aware framework* that localizes audio-critical moments, generates visual keyframes accordingly, and interpolates intermediate frames. This selective temporal focus enables smoother motion transitions and sharper audio-visual synchronization without the overhead of dense uniform sampling.

**Keyframe-based Video Processing.** In video processing, keyframes are pivotal in compressing video clips by retaining essential features, thereby facilitating efficient analysis of lengthy videos or high-dynamic motions (Kulhare et al., 2016; Shen et al., 2024; Lee et al., 2024; Xu et al., 2024; Ataallah et al., 2024). In the realm of video generation, keyframes serve as foundational references, enabling the synthesis of intermediate frames that ensure temporal coherence and visual consistency. For long video generation, current approaches employ keyframe-based generation pipelines to enhance long-term coherence in video synthesis (Zheng et al., 2024; Yin et al., 2023). Others focus on interpolation techniques from keyframes, which predict missing frames between keyframes input, ensuring motion realism and visual consistency in dynamical motions (Geng et al., 2024).

## 3 Methods

In this section, we present our keyframe-aware audio-conditioned video generation framework **KeyVID**. Given an input audio and the first frame of a video, we follow a three-stage generation process (Fig. 1(b)) and train three separate models: (1) **Keyframe Localizer** predicts a motion score curve from the input audio and detects the keyframe positions (Sec. 3.1); (2) **Keyframe Generator** generates keyframe images at detected keyframe positions conditioned on the input image and audio (Sec. 3.2); (3) **Motion Interpo-**

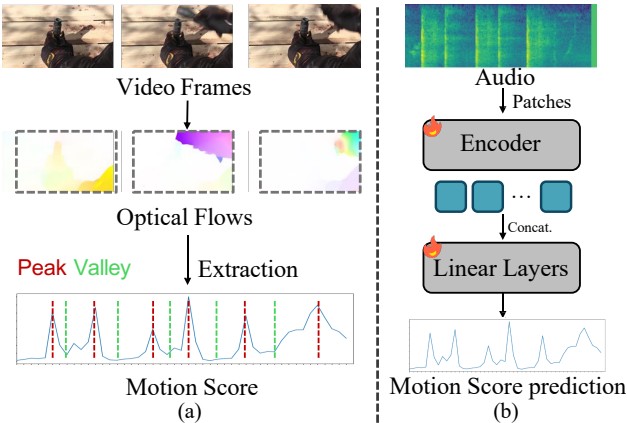

Figure 2: **Motion score computation and prediction.** (a) We compute motion scores as the average of the optical flow of each frame and localize keyframe from the peaks and valleys. (b) Keyframe localizer is trained to predict motion scores from audio to identify keyframe locations.

**lator** synthesizes intermediate frames to reconstruct a smooth video with dense frames conditioned on the generated keyframe images and input audio (Sec. 3.3).

### 3.1 Keyframe Localization from Audio

We train a keyframe localizer to infer keyframe locations from input by exploiting the correlation between acoustic events and motion changes. For instance, a hammer striking a table generates a sharp sound that often aligns with a sudden visual transition. The network learns to predict motion scores from the input audido and then localizes keyframes from the motion score sequence.

**Optical Flow based Motion Score.** To train the keyframe localizer, we first generate keyframe labels by analyzing optical flow from training video sequences, as shown in Fig. 2(a). We first obtain a *motion score* for each frame by calculating the optical flow and averaging it across all pixels to represent the motion intensity of the frame. These scores collectively form a temporal motion curve across the frames.

Specifically, we employ a pre-trained RAFT model (Teed & Deng, 2020) as the optical flow estimator. Given a video clip consisting of frames $\{I_j\}_{j=1}^T$, RAFT computes the optical flow field $\mathbf{OF}_t$ between two frames $I_j$ and $I_{j+1}$. The optical field consists of horizontal ($u_t$) and vertical ($v_t$) components at each pixel, and the motion score $M(t)$ of frame $t$ is calculated as:

$$M(t) = \sum_{i,j} \left( |u_t(h,w)| + |v_t(h,w)| \right), \tag{1}$$

where $t = 1, \ldots, T-1$ denotes the time step of the video with $T$ frames. $(h, w)$ represents the pixel location.

**Motion Score Prediction.** We train the keyframe localizer to predict motion scores from input audio, enabling it to learn the underlying relationship between motion dynamics and acoustic cues. As shown in Fig. 2(b), the keyframe localizer first converts the raw audio into a spectrogram and extract audio features using a pretrained Transformer-based encoder (Girdhar et al., 2023). To better align the audio features with the temporal resolution of motion cues, we modify the patchify stride to increase the number of patches and interpolate the positional embeddings of the encoder (see Appendix A). The audio features are then passed through fully connected layers to predict motion scores. We train the model with $\mathcal{L}_1$ loss between the prediction and the ground-truth motion score calculated by Eq. (1).

**Keyframe Selection.** Given motion scores $\{M(t)\}_{t=1}^T$ of the video frames, we select $T_K \ll T$ keyframes that capture salient motion dynamics with minimal redundancy. Keyframes are identified from local maxima ("peaks") and minima ("valleys"), which indicate dramatic motion changes (Wolf, 1996; Kulhare et al., 2016). We first include the initial frame and sample up to $\frac{T_K}{2} - 1$ peaks; if fewer peaks exist, all are used. For each pair of peaks, we select one valley to preserve motion completeness. The remaining keyframes are obtained

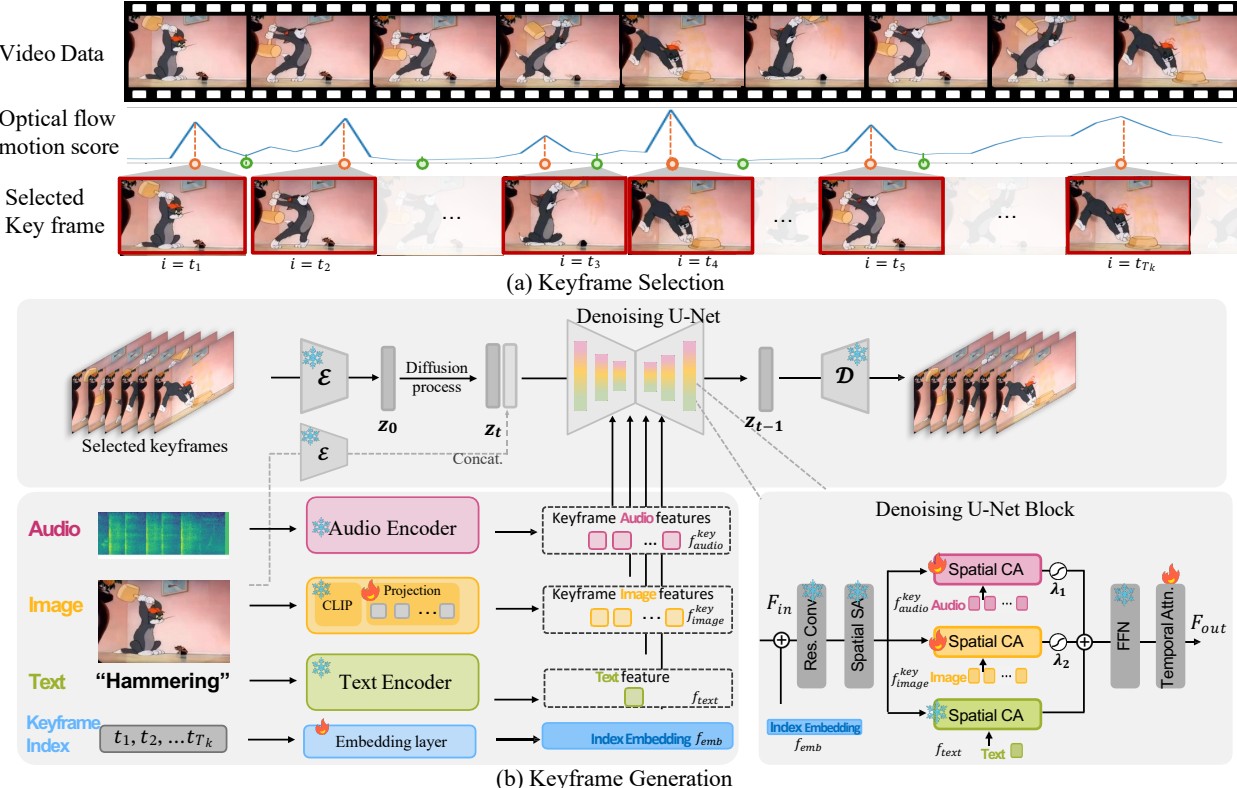

Figure 3: **Keyframe data selection and keyframe generator**. (a) We select keyframes based on the local maxima and minima of the motion score. (b) The keyframe generator is trained to generate these sparse keyframes conditioned on the audios, first frame image, text, and keyframe indices. These conditions are encoded and passed into the denoising U-Net. In each denoising U-Net block, the index embeddings are added with video features and passed into Residual convolutional block (**Res. Conv.**). The following layers contain a spatial self-attention (**SA**) and spatial cross attention (**CA**) on each three conditional features. The output of each CA is followed by a gating with learnable weights $\lambda_1$ and $\lambda_2$. Please see details in Sec. 3.2.

by evenly sampling across frame bins. This design ensures robustness to sequences with smooth motion or weak audio cues. Further details and examples are provided in Appendix A and D. We use the selected $T_K$ keyframes to train the keyframe generator and the keyframe indices $\{t_i\}_{i=1}^{T_K}$ serve as additional input conditions.

## 3.2 Audio-conditioned Keyframe Generation

We propose a novel keyframe generator network to generate $T_K$ keyframes for a video sequence of length $T$ from the input audio and first frame image. Unlike previous video generation models (Xing et al., 2024; Zhang et al., 2024b) that are trained on uniformly downsampled frames, the keyframe generator aims to generate sparse keyframes that captures crucial motions. To enable this, we propose two key designs: (1) *Frame Index Conditioning* - we introduce keyframe index embedding that encodes each frame's absolute position, which provides explicit temporal anchors and ensures coherence when generating non-uniformly distributed frames; (2) *Keyframe-aligned Feature Extraction* - we extract image and audio features that are aligned with the corresponding keyframe time steps to serve as accurate conditions for keyframe generation. In the following, we first provide an overview of the keyframe generator and explain the input conditioning in details.

**Overview.** We leverage the image dynamic prior of pretrained text-to-video latent diffusion models, and inject the input audio, first frame, and keyframe indices as additional input conditions. The model architecture is shown in Fig. 3(b). We encode the selected keyframes into a latent code $\mathbf{z_0} \in \mathbb{R}^{T_k \times C \times H \times W}$

with a pretrained encoder $\mathcal{E}$, where $H$ and $W$ denotes the spatial dimensions, and $C$ denotes the feature channels. The denoising U-Net learns to iteratively denoise the noisy latent code $\mathbf{z_t}$, and the input conditions are encoded and injected into each denoising U-Net block. The final keyframes are generated from the denoised latent code using the pretrained decoder $\mathcal{D}$.

**Frame Index Embedding.** Off-the-shelf video diffusion models assume uniformly sampled frames and cannot directly handle sparsely distributed keyframes. To address this, we introduce a frame index embedding layer that encodes the absolute index of each keyframe $\{t_i\}_{i=1}^{T_K}$ within the original video sequence into frame index embedding $\mathbf{f}_{\mathrm{emb}} \in \mathbb{R}^{T_K \times C}$. $\mathbf{f}_{\mathrm{emb}}$ is added with the latent video features $\mathbf{z}$ before passing into the denoising U-Net blocks, ensuring explicit positional information is provided to the network for global temporal consistency and accurate cross-modal alignment.

**Audio Feature Condition.** We use a pretrained ImageBind audio encoder (Girdhar et al., 2023) to extract audio features for video synthesis. Given an input spectrogram $\mathbf{A} \in \mathbb{R}^{C_A \times T_A}$, the encoder splits it into overlapping patches of size $(c_a, t_a)$ with a stride $\Delta t < t_a$ and encodes it into a sequence of feature embeddings $\{\mathbf{h}_i\}_{i=1}^N$ using Transformer layers. We decrease the patchify stride $\Delta t$ of the pretrained encoder to obtain finer-grained temporal embeddings. We segment the extracted audio features into $T$ time steps to match the full video length, resulting in $\mathbf{f}_{\mathrm{audio}} \in \mathbb{R}^{T \times C \times M}$, where $M$ is the number of audio features in each time step. Using the keyframe indices $\{i_t\}_{t=1}^{T_K}$, we extract the corresponding $T_K$ audio features from the full $T$-length sequence and obtain the keyframe-aligned audio features $\mathbf{f}_{\mathrm{audio}}^{\mathrm{key}} = \{\mathbf{f}_{\mathrm{audio}}^{(i_t)}\}_{t=1}^{T_K}$. These keyframe-aligned audio features are fused with text and image conditions via cross-attention layers in the U-Net, ensuring accurate synchronization between generated keyframes and their associated audio cues.

**Image Feature Condition.** The first frame image $\mathbf{I}$ is injected into the keyframe generation process via two pathways. First, we extract the image feature using a frozen CLIP image encoder (Radford et al., 2021). We project the image features into $T$ frame-specific image conditions using a Q-Former Li et al. (2023a) projection layer, yielding $\mathbf{f}_{\mathrm{img}} \in \mathbb{R}^{T \times C \times H \times W}$. We then select the corresponding $T_K$ features using keyframe indices $\{i_t\}_{t=1}^{T_K}$ to obtain keyframe-aligned image feature $\mathbf{f}_{\mathrm{img}}^{\mathrm{key}} \in \mathbb{R}^{T_K \times C \times H \times W}$. Second, we encode the image with the encoder $\mathcal{E}$, concatenate it with noisy latent code $\boldsymbol{z}_t$, and feed them to the denoising U-Net. This provides additional visual details from $\mathbf{I}$ to guide the keyframe generation (Xing et al., 2024).

**Text Feature Condition.** Following prior work, we encode the text prompt of the video using a frozen CLIP text encoder ((Radford et al., 2021). The extracted text embedding $\mathbf{f}_{\mathrm{text}}$ is repeated for all $T_K$ keyframe to provide consistent semantic guidance during the denoising process.

**Feature Fusion.** Each conditioning feature ($\mathbf{f}_{\mathrm{audio}}^{\mathrm{key}}$, $\mathbf{f}_{\mathrm{img}}^{\mathrm{key}}$, and $\mathbf{f}_{\mathrm{text}}$) is processed separately through spatial cross-attention layers in the U-Net blocks. Given input latent features $\mathbf{F}_{\mathrm{in}}$, we compute query projections $\mathbf{Q} = \mathbf{F}_{\mathrm{in}} \mathbf{W}_Q$ and apply spatial attention to text, image, and audio features:

$$\mathbf{F}_{\mathrm{out}} = \mathrm{SA}(\mathbf{Q}, \mathbf{K}_{\mathrm{text}}, \mathbf{V}_{\mathrm{text}}) + \lambda_1 \cdot \mathrm{SA}(\mathbf{Q}, \mathbf{K}_{\mathrm{audio}}, \mathbf{V}_{\mathrm{audio}}) + \lambda_2 \cdot \mathrm{SA}(\mathbf{Q}, \mathbf{K}_{\mathrm{img}}, \mathbf{V}_{\mathrm{img}}). \tag{2}$$

where SA stands for spatial attention, $\mathbf{K}$ and $\mathbf{V}$ are the key and value projections for each modality, and $\lambda_1$, $\lambda_2$ are learnable fusion weights. The fused features are then processed through a feedforward network (FFN) and temporal self-attention to ensure spatial and temporal consistency.

## 3.3 Motion Interpolation

After generating $T_K$ keyframes, we use a *motion interpolator* to generate the missing frames to obtain the a full video sequence of length $T$. Interpolation has been widely used in uniform frame generation (Blattmann et al., 2023a; Xing et al., 2024), where a model predicts a fixed number of intermediate frames given the first and last frame. However, for keyframe-based generation, the positions of missing and available frames vary, introducing additional challenges. To address this, we adapt our *keyframe generator* diffusion model into a *motion interpolator* model that generates $T_K$ frames at once using masked frame conditioning. The overall architecture remains mostly the same, with the primary difference in how image conditions are incorporated. Rather than conditioning solely on the first frame, the model utilizes the features of generated keyframes as conditions, thereby learning to synthesize the missing frames in between. This approach facilitates interpolation between non-uniformly distributed keyframes while maintaining temporal consistency. Details can be

Table 1: Performance on the *AVSync15* and the *Greatest Hits* datasets. Best is marked in **bold**.

| Input | Model | AVSync15 | | | | | | Greatest Hits | | | | | |
|---|---|---|---|---|---|---|---|---|---|---|---|---|---|
| | | FID↓ | IA↑ | IT↑ | FVD↓ | AlignSync↑ | RelSync↑ | FID↓ | IA↑ | IT↑ | FVD↓ | AlignSync↑ | RelSync↑ |
| T+A | TPoS | 13.5 | 23.38 | 24.83 | 2671.0 | 19.52 | 42.50 | 33.85 | 11.50 | **17.90** | 3327.90 | 21.48 | 44.90 |
| | TempoToken | 12.2 | 18.84 | 17.45 | 4466.4 | 19.74 | 44.05 | 25.90 | 4.88 | 9.28 | 3300.53 | 21.56 | 45.38 |
| I+T | I2VD | 12.1 | - | 30.35 | 398.2 | 21.80 | 43.92 | 9.10 | - | 13.42 | 425.0 | 22.05 | 44.58 |
| | DynamiCrafter | 11.7 | - | 30.02 | 400.7 | 21.76 | 43.68 | 12.40 | - | 13.73 | 337.71 | 22.82 | 45.85 |
| I+T+A | CoDi | 14.5 | 28.15 | 23.42 | 1522.6 | 19.54 | 41.51 | 21.78 | 12.01 | 14.11 | 1336.00 | 22.30 | 45.35 |
| | TPoS | 11.9 | 38.36 | **30.73** | 1227.8 | 19.67 | 39.62 | 28.43 | 9.36 | 13.19 | 1370.57 | 22.04 | 45.55 |
| | AADiff | 18.8 | 34.23 | 28.97 | 978.0 | 22.11 | 45.48 | - | - | - | - | - | - |
| | AVSyncD | 11.7 | 38.53 | 30.45 | 349.1 | 22.62 | 45.52 | **8.70** | 12.07 | 13.31 | 249.30 | 22.83 | 45.95 |
| | **KeyVID (Ours)** | **11.1** | **39.21** | 30.12 | **263.3** | **24.44** | **49.06** | 12.10 | **12.40** | 15.66 | **202.10** | **22.91** | **46.03** |
| | Static | - | 39.76 | 30.39 | 1220.4 | 21.83 | 43.66 | - | 13.33 | 16.56 | 348.9 | 24.36 | 48.73 |
| | Groundtruth | - | 40.06 | 30.31 | - | 25.04 | 50.00 | - | 13.52 | 16.49 | - | 25.02 | 50.00 |

found in Appendix C. To generate a full video with $T$ frames in a single pass, we incorporate FreeNoise (Qiu et al., 2023) to increase the number of output frames during inference. This allows the interpolation model to take all generated keyframes as conditioning inputs and predict all missing frames in one single step. Further details on the training and inference time of this model are provided in the Appendix G.

# 4 Experiments

## 4.1 Implementation Details

**Datasets.** We train and evaluate our method on three datasets: *AVSync15* (Zhang et al., 2024b), *Greatest Hits* (Owens et al., 2016), and *Landscapes* (Lee et al., 2022). *AVSync15* is a subset of the VGG-Sound (Chen et al., 2020) dataset, consisting of fifteen classes of activities with highly synchronized audio and video captured in the wild. Some activities have more intense motions, such as hammer hitting and capgun shooting. *Greatest Hits* contains videos of humans hitting various objects with a drumstick, producing hitting sounds that are temporally aligned with the motions. *Landscapes* is a collection of natural environment videos with corresponding ambient sounds without synchronized video motion. We sample two-second audio-video pairs from these datasets for experiments. Videos were sampled at 24 fps with 48 frames, and resized to $320 \times 512$. Audios were sampled at 16kHz and converted into 128-d spectrograms. We set $T_K = 12$ as the temporal length of keyframe generation and interpolations.

**Training.** We adopted the pre-trained DynamiCrafter (Xing et al., 2024) as the backbone video diffusion model and pre-trained ImageBind (Girdhar et al., 2023) as the audio encoder. All models were trained using Adam optimizer with a batch size of 64 and a learning rate of $1 \times 10^{-5}$.

**Baselines.** We follow (Zhang et al., 2024b) to compare our method with the simple *static* baseline where the input frame is repeated to form a video, as well as state-of-the-art video generation models with different input modalities: **(1) T+A** is the video generation model conditioned only on text and audio, such as TPoS (Jeong et al., 2023) and TempoToken (Yariv et al., 2024). **(2) I+T** includes many state-of-the-art video generation models, which are conditioned on images and text prompts. We compare with I2VD (Zhang et al., 2024b), VideoCrafter (Chen et al., 2023a) and DynamiCrafter (Xing et al., 2024). **(3) I+T+A** takes image, text and audio inputs for video generation, which includes CoDi (Tang et al., 2023b), TPoS (Jeong et al., 2023), AADiff (Lee et al., 2023) and AVSyncD (Zhang et al., 2024b).

**Metrics**. We use the Frechet Image Distance (**FID**) (Heusel et al., 2017) and Frechet Video Distance (**FVD**) (Unterthiner et al., 2018) to evaluate the visual quality of the individual frames and videos. We also compare the average image-text (**IT**) and image-audio (**IA**) semantic alignment scores of video frames using CLIP (Radford et al., 2021) and ImageBind (Girdhar et al., 2023). To measure audio-video synchronization, we evaluate the generated videos with **RelSync** and **AlignSync** proposed by Zhang et al. (2024b).

## 4.2 Quantitative Results

Table 1 presents the quantitative evaluation results on the *AVSync15* and *Greatest Hits* datasets. Results on the *Landscape* dataset can be found in the Appendix J. On the *AVSync15* dataset, KeyVID demonstrates

Table 2: Ablation study results on *AVSync15*.

| Setting | FID↓ | FVD↓ | AlignSync↑ | RelSync↑ |
|---|---|---|---|---|
| **KeyVID** | 11.1 | 263.3 | **24.44** | **49.06** |
| **KeyVID-Uniform** | **11.0** | 273.4 | 23.53 | 47.23 |
| | (-0.9%) | (+3.8%) | (-3.7%) | (-3.7%) |
| w/o *Frame Index* | **11.0** | **258.9** | 23.93 | 47.90 |
| | (-0.9%) | (-1.7%) | (-2.1%) | (-2.4%) |
| w/o *First Frame* | 11.7 | 265.5 | 24.02 | 48.49 |
| | (+5.4%) | (+0.8%) | (-1.7%) | (-1.2%) |

superior performance across both audio-visual synchronization and visual quality metrics. It achieves the highest synchronization scores with AlignSync of 24.44 and RelSync of 49.06, substantially outperforming the previous state-of-the-art AVSyncD (22.62 and 45.52, respectively). These improvements highlight the effectiveness of our keyframe-aware strategy in capturing critical dynamic moments that align with audio events. In terms of visual quality, KeyVID also excels with an FID score of 11.00 and FVD score of 263.3, representing the best performance among all compared methods. Additionally, our approach achieves the highest image-audio semantic alignment score (IA: 39.21), demonstrating strong correspondence between generated visual content and audio input. The *Greatest Hits* dataset presents a particularly challenging scenario with distinct percussive audio events that require precise temporal alignment with visual motions. KeyVID achieves competitive performance across all evaluation metrics. Notably, KeyVID attains the best FVD score of 202.10, indicating superior visual quality in the generated videos. For audio-visual synchronization, KeyVID achieves AlignSync and RelSync scores of 22.91 and 46.03, respectively, outperforming most baseline methods while maintaining strong visual quality with competitive FID performance.

## 4.3 Ablation Study

**Keyframe vs. Uniform Sampling.** To validate the effectiveness of keyframe-aware generation, we compare KeyVID with a uniform sampling baseline, **KeyVID-Uniform**, where KeyVID-Uniform generates 12 uniform frames instead of keyframes before motion interpolation. As shown in Table 2, KeyVID consistently outperforms KeyVID-Uniform across all metrics,

with larger improvements in audio-visual synchronization scores AlignSync and RelSync, while maintaining competitive visual quality metrics. In addition, KeyVID achieves greater improvement in high-intensity motion scenarios as shown in Fig. 5. These results confirm our hypothesis that strategically selecting keyframes based on audio and motion cues leads to superior audio-visual synchronization.

**Frame Conditioning.** We further analyze the contribution of two components in our frame conditioning mechanism in Table 2. Removing the frame index embedding leads to degraded audio-visual synchronization, with AlignSync and RelSync scores decreasing by 2.1% and 2.4%, respectively. This demonstrates that frame index embedding provides crucial temporal information that helps the model understand the sequential ordering of keyframes during generation.

Removing the first-frame condition from the motion interpolator results in significant performance degradation, particularly in visual quality metrics. The FID increases by 5.4% and FVD increases by 0.80%, indicating that the first frame serves as an essential reference for maintaining visual consistency during interpolation. The combination of both components achieves optimal performance, confirming the importance of our complete frame conditioning design.

## 4.4 Visualization

Fig. 4 presents qualitative comparisons between KeyVID and baseline approaches on different type of motions. Our keyframe-aware approach more accurately captures motion peaks that align with audio events, such as the exact moment of impact in hammering or the smoke in gun shooting. Compared to the uniform frame sampling variant KeyVID-Uniform, KeyVID better preserves temporal coherence by focusing on key moments of motion. In sequences like dog barking and frog croaking, KeyVID ensures that mouth

movements align precisely with sound peaks, whereas KeyVID-Uniform and AVSyncD introduce temporal misalignment or missing frames. For subtle motions, such as playing the trombone or violin, our model still produces smooth and stable movements, even during sustained notes or brief pauses in the audio, where motion cues are weak and baselines tend to jitter or freeze. Additional video visualizations for intensive, moderate, and subtle motions are provided in the supplementary material.

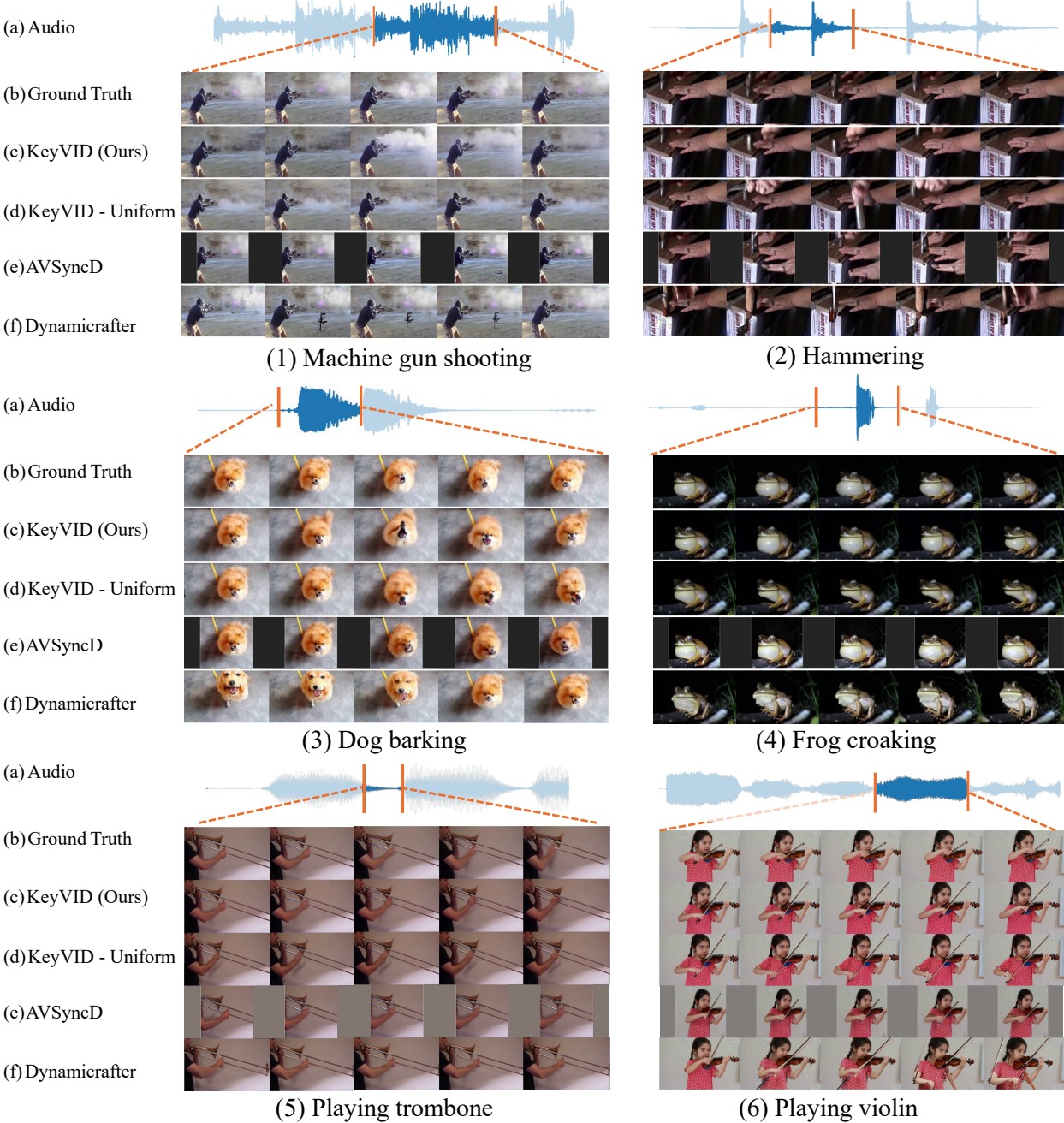

Figure 4: **Qualitative comparison of KeyVID and baseline methods**. We crop key motions on the audio waveform in (a) and the corresponding ground truth video in (b) as references and compare the generated video clips between models from (c) to (f). KeyVID with keyframe awareness (c) shows better alignment with motion peaks in audio signals—for example, the hammer striking, gunshots producing smoke, or facial movements when dogs bark or frogs croak. For subtle motion scenarios such as playing the trombone or violin, our model is also able to produce smooth, stable motion during the brief pauses or sustained notes.

### 4.5 Effects of Motion Intensity

To analyze how KeyVID performs across different motion types, we categorize the 15 classes in the AVSync15 dataset into three intensity levels based on their average motion scores: *Subtle*, *Moderate*, and *Intense*, with five classes each. The *Intense* level includes highly dynamic motions such as hammering and dog barking, while the *Subtle* level consists of activities with slow movement, such as playing the violin or trumpet. Fig. 5 compares RelSync scores across these motion intensities for KeyVID, KeyVID-Uniform, and AVSyncD. KeyVID shows increasing improvements over KeyVID-Uniform as motion intensity rises, with RelSync gains of 1.50, 1.59, and 2.01 for *Subtle*, *Moderate*, and *Intense* motions, respectively. This demonstrates the effectiveness of keyframes in capturing audio rapid motion transitions Compared to AVSyncD, KeyVID consistently achieves superior synchronization with RelSync gains of 3.86, 3.18, and 3.07 across all intensity levels.

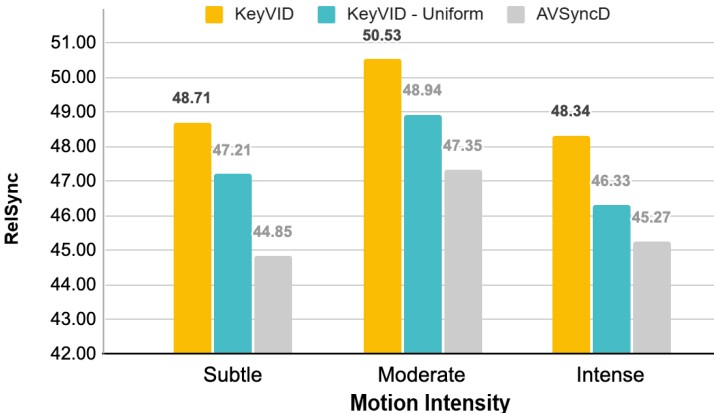

Figure 5: **RelSync scores across motion intensity levels**. **KeyVID** improves audio synchronization score on all motion intensity.

### 4.6 User Study

We conducted a user study with twelve participants to assess the quality of generated videos. Each participant was shown twenty randomly selected video samples, where each sample contained results from four models presented in a random order with the same inputs. They were asked to choose which video exhibited better audio-visual synchronization, visual quality, and temporal consistency. We aggregated all $12 \times 20 = 240$ votes for each metric and computed the percentage of votes each model received, as shown in Tab. 3. Further details on the user study can be found in Appendix F.

Table 3: **User study results**. Participants voted for the best method based on audio synchronization (**AS**), visual quality (**VQ**), and temporal consistency (**TC**). The numbers represent the percentage of votes each model received for each metric.

| Models | AS | VQ | TC |
|---|---|---|---|
| **KeyVID** | **66.25%** | **65.00%** | **65.00%** |
| **KeyVID-Uniform** | 17.92% | 22.08% | 21.67% |
| AVSyncD | 11.67% | 7.08% | 7.92% |
| DynamiCrafter | 4.17% | 5.83% | 5.42% |

### 4.7 Open-Domain Audio-Synchronized Visual Animation

We show KeyVID's ability to animate open-domain inputs beyond its training distribution. As illustrated in Fig. 6, we use the first frame from a Sora-generated video clip, where a hammer is held in the air before striking down. We control the visual animation through two distinct hammering audio clips: the first

contains metallic strike sounds, while the second captures impacts on a wooden surface. Our model not only successfully generates videos that match the temporal pattern of strikes, but also adapts the motion based on the material properties inferred from the audio: the first video shows hammering on metal nails, while the second shows hammering on a wooden table. These results demonstrate the generalization capability of KeyVID to open-domain inputs and its ability to accurately follow the audio semantics for visual animation.

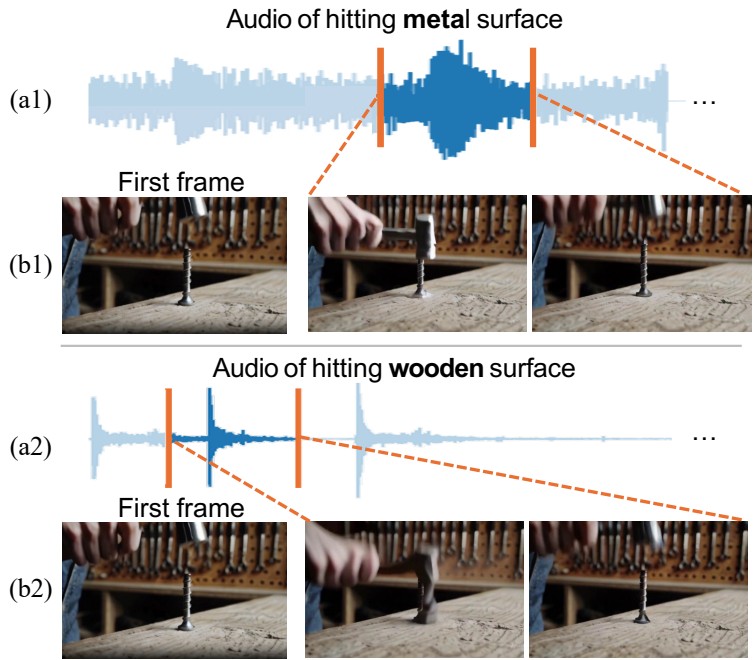

Figure 6: **Open-domain video generation**. Given the same first frame and different audio inputs (a1) and (a2), KeyVID synthesizes videos that align with the audio's semantic meaning and motion pattern in (b1) and (b2).

## 5 Conclusion

In this paper, we introduced a keyframe-aware audio-synchronized visual animation model which enhances video generation quality and audio alignment, particularly for highly dynamic motions. Our approach first localizes keyframes from audio and generates corresponding frames using a diffusion model. Then we synthesize intermediate frames to obtain smooth high-frame-rate videos while maintaining memory efficiency. Experimental results demonstrate superior performance across multiple datasets, especially in scenarios with intensive motion. Compared to previous methods, our model significantly improves audio-visual synchronization and visual quality.

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
