# OpenReview forum: "KeyVID: Keyframe-Aware Video Diffusion for Audio-Synchronized Visual Animation"
_TMLR — Rejected by TMLR_

### Review · Reviewer_enZT · 2026-03-16

**Summary Of Contributions:**

This paper proposes KeyVID, a keyframe-aware diffusion framework for audio-synchronized video generation. The goal is to generate videos whose motions are temporally aligned with input audio, addressing the limitation of existing methods that rely on uniformly sampled frames and often miss important audio-visual events. The key idea is to first predict keyframe positions from audio, generate only these sparse keyframes with a diffusion model, and then interpolate intermediate frames to produce a full video. The novelty lies in introducing audio-driven keyframe localization and sparse keyframe-based video generation, which focuses model capacity on important motion moments instead of generating all frames uniformly. Experiments show that the method improves audio–video synchronization and visual quality, outperforming prior approaches on benchmarks such as AVSync15.

Strengths:

1. The paper introduces a keyframe-aware generation paradigm, which is a simple but effective insight: generate only important frames and interpolate the rest instead of uniformly generating all frames. By predicting keyframe positions from audio, the model focuses on moments where motion is likely to happen, improving alignment between sound and visual actions.

2. The proposed method shows clear improvements in synchronization metrics and video quality compared with prior approaches on multiple datasets.


Weaknesses:

A potential limitation of the proposed approach is that it implicitly assumes a strong correlation between audio events and visual motion. The ground-truth keyframes are derived from motion peaks computed using optical flow in the video, while during inference the model predicts these motion peaks solely from audio signals. This design relies on the assumption that important visual motions are typically accompanied by corresponding sounds. However, in more general scenarios this assumption may not always hold. For example, some visual motions may occur without producing noticeable audio, or multiple sound sources may exist in a scene that are not directly associated with visible objects. In such cases, the predicted keyframes may not accurately reflect the true visual dynamics, potentially limiting the robustness and generalization of the method.

**Additional Comments:**

If multiple objects in the scene produce sounds simultaneously, can the model generate a corresponding video that reflects these audio sources correctly?

**Audience:**

Yes

**Audience Explanation:**

Researchers in TMLR working on generative models, video diffusion, and multimodal learning may be interested in this paper. The work introduces a keyframe-aware approach for audio-conditioned video generation, which offers a new perspective on improving audio–visual synchronization and efficient temporal modeling in video generation.

**Broader Impact Concerns:**

A potential ethical concern is that audio-driven video generation models could be misused to create misleading or manipulated videos that appear synchronized with real audio events. As such technologies improve in realism, they may contribute to challenges in verifying the authenticity of visual media. Therefore, the authors should consider discussing potential misuse and responsible deployment in a Broader Impact Statement.

**Claims And Evidence:**

Yes

**Claims Explanation:**

The authors present a large number of quantitative and qualitative results and provide comparison videos, demonstrating that the proposed method can generate videos with better audio–visual synchronization.

**Requested Changes:**

The current paper fixes the number of keyframes T for all experiments, but it does not analyze how the number of keyframes affects generation quality and audio–visual synchronization. Since the proposed framework relies on sparse keyframe generation followed by interpolation, the choice of  T is likely an important factor that influences both motion fidelity and computational cost. It would strengthen the paper if the authors could include an ablation study evaluating different numbers of keyframes.

---

### Review · Reviewer_ZGgT · 2026-03-21

**Summary Of Contributions:**

This paper proposes KeyVID, a keyframe-aware framework for audio-synchronized visual animation. The core idea is to avoid uniformly sampling sparse frames for training/generation, since uniform sampling can miss salient audio-visual events in highly dynamic clips. Instead, the method first predicts motion-score peaks from audio, uses those predicted positions to generate sparse but informative visual keyframes, and then interpolates the remaining frames to reconstruct a high-frame-rate video. The full system has three components: a keyframe localizer trained from optical-flow-derived motion scores, a keyframe generator conditioned on audio, first-frame image, text, and frame-index embeddings, and a motion interpolator that fills in the missing frames. The paper reports improvements over prior baselines on AVSync15 and Greatest Hits, especially on synchronization metrics, and includes ablations, qualitative comparisons, motion-intensity analysis, and a small user study.

**Audience:**

Yes

**Audience Explanation:**

Yes. This paper sits at the intersection of multimodal generation, video diffusion, and audio-visual learning, all of which are active areas of interest to the TMLR audience. The problem is well motivated: generating videos that are not only visually plausible but also temporally synchronized with audio is an important challenge, and the paper focuses on a specific failure mode of existing approaches. The proposed keyframe-aware perspective is a useful contribution because it reframes synchronization as selective temporal modeling rather than simply increasing frame rate or using uniform sparse sampling. Researchers working on video generation, multimodal diffusion, audio-visual alignment, and efficient temporal modeling would likely find the approach and the empirical findings relevant.

**Broader Impact Concerns:**

The work appears to be primarily a technical contribution on controllable video generation, and I do not see a unique ethical concern beyond those already associated with modern generative video systems.

**Claims And Evidence:**

Yes

**Claims Explanation:**

Overall, the empirical evidence supports the paper’s main claims, though not perfectly completely.

The paper’s central claim is that keyframe-aware generation improves audio-visual synchronization relative to uniform sparse sampling and prior audio-conditioned baselines. This is reasonably supported by the quantitative results. On AVSync15, KeyVID improves over AVSyncD on both synchronization metrics (AlignSync 24.44 vs. 22.62, RelSync 49.06 vs. 45.52) and also improves FVD (263.3 vs. 349.1), which is a strong result because it suggests the synchronization gains do not come at the expense of video quality. The ablation against the uniform variant also directly supports the key methodological claim: KeyVID outperforms KeyVID-Uniform on FVD, AlignSync, and RelSync.

Some limitations remain. The improvements are strongest on AVSync15 and less dramatic on Greatest Hits. The user study uses only 12 participants and 20 samples per participant, which is useful but not especially large. The paper also does not deeply analyze cases where the audio-to-motion correlation may be weak or ambiguous, and there is limited discussion of the cost/latency tradeoff of the three-stage pipeline compared with simpler baselines. So the evidence is convincing enough for the main claims, but some claims around generality and efficiency would benefit from stronger substantiation.

**Requested Changes:**

1. The paper should provide a clearer accounting of computational cost and efficiency. Since the method uses three stages (localizer, keyframe generator, interpolator), the reader needs a more transparent comparison of training/inference cost, memory, and runtime versus AVSyncD and the uniform variant. The paper motivates the method partly by efficiency relative to high-frame-rate training, but the current main text does not quantify this clearly enough.

2. The paper should discuss failure cases and boundary conditions more explicitly. In particular, it would be helpful to understand when audio-predicted keyframes are inaccurate, when audio events do not map cleanly to visible motion, and whether the method degrades on clips with weak or diffuse synchronization cues, such as more ambient scenes. The Landscapes dataset is mentioned, but the main paper does not analyze this regime in depth.

3. The authors should better justify and describe the keyframe-selection heuristic. The optical-flow-based motion score and peak/valley extraction are intuitive, but the design choices are somewhat heuristic. A stronger explanation of why these labels are appropriate supervision for audio-based localization, along with sensitivity to the number of keyframes or the peak/valley selection rule, would improve confidence in the approach.

---

### Review · Reviewer_vsMf · 2026-04-15

**Summary Of Contributions:**

This paper proposes a method for generating keyframes by predicting moments of highly dynamic motion directly from the audio. These predicted keyframes are then used as anchors, with interpolation filling in the remaining frames to produce a complete video. Experimental results on the AVSync15 and Greatest Hits datasets show that the approach achieves improved audio–visual alignment compared to existing methods.

**Audience:**

Yes

**Audience Explanation:**

Audio-to-video (A2V) generation is currently a very active area in the AI community, and this paper tackles an important aspect of it: audio–video alignment. By focusing on this problem, the work has the potential to benefit future A2V research. Overall, I think many researchers in the community would find this paper both relevant and interesting.

**Broader Impact Concerns:**

The potential impact of this work is broadly applicable to video generation tasks in general. At this stage, I don't see any specific concerns that are unique to this paper beyond those already associated with video generation methods as a whole.

**Claims And Evidence:**

No

**Claims Explanation:**

The main contributions of the paper are twofold. First, the authors argue that generating keyframes explicitly can lead to better overall results. Second, they show that audio alone can be used to predict these keyframes.

For the first point, the paper does a solid job of explaining the impact of keyframe generation. Table 1 clearly illustrates how the approach improves performance, while Table 2 breaks down the contribution of each component. The authors also go a step further by analyzing how the method performs across different types of motion, which helps make the results more convincing.

As for the second point, the authors provide additional evidence in Appendix D, where they present results supporting the idea that audio can effectively guide keyframe prediction.

**Requested Changes:**

Overall, I found the paper to be well-written and easy to follow. Since I’m not an expert in the A2V field, I can't say with full confidence whether any important baselines are missing, but from my perspective, the experimental setup feels reasonable and sufficient. That said, I do have a few suggestions that could further strengthen the paper:

* It would be helpful to move at least part of the motion prediction results (currently in Appendix D) into the main paper. Right now, this aspect isn't really discussed in the main text, which makes it harder for readers to understand what the predicted motion actually looks like. Even a small subset would make a big difference, with the rest remaining in the appendix.
* I would also encourage the authors to include a more in-depth analysis of the motion prediction results. For instance, how does the method perform in more complex audio scenarios, such as multi-channel sound or the presence of background music? Does it remain robust, or does performance degrade? Additionally, while some failure cases are shown on the project website, it would be useful to explicitly include and discuss them in the paper (either in the main text or appendix).
* In the ablation study, the *w/o Frame Index* setting appears to achieve slightly better FID and notably better FVD. This is quite interesting, does it suggest that the model’s prior is not well-aligned with the use of frame index information? It might be worth discussing this in more detail. For example, instead of absolute positional embeddings, would relative positional encoding work better here, given its popularity in modern generative models? Exploring this direction could potentially lead to further improvements.

---

### Decision · Action_Editor_mSzw · 2026-06-09

**Recommendation:** Reject

**Audience:**

Yes

**Audience Explanation:**

The TMLR community would be interested in the problem of audio-visual synchronization and keyframe-aware video generation.

**Claims And Evidence:**

No

**Claims Explanation:**

While the initial submission provides some promising experimental results, the authors did not submit a rebuttal or address the reviewers' concerns regarding the method's underlying assumptions, necessary ablation studies, and failure cases. Therefore, without this critical response, the claims currently lack the necessary justification and robustness expected for publication.

---

> ### Author Response · Authors · 2026-06-09
> **Request for a Late Rebuttal Submission**
>
> Dear Action Editor,
>
> We sincerely apologize for the lack of a timely response. We are very eager to address the reviewers' valuable feedback and have already begun preparing our revisions.
>
> Unfortunately, the email with the subject "[TMLR] Reviewer responses and discussion for your TMLR submission" did not reach the primary author responsible for preparing the rebuttal, and we only became aware of the update after noticing it in the OpenReview notification. As a result, we were unaware that this phase had begun and that a rebuttal deadline had been set.
>
> We sincerely apologize for missing the deadline. Given the circumstances, would it be possible to grant us a brief exception to submit our rebuttal and revised manuscript later this week? We would greatly appreciate the opportunity to address the reviewers' concerns and improve the paper.
>
> If this is not possible within the current cycle, we completely understand. Thank you very much for your time and consideration.
>
> Best regards,
>
> The Authors